# Automatic Fuzzy Logic-Based Maize Common Rust Disease Severity Predictions with Thresholding and Deep Learning

**DOI:** 10.3390/pathogens10020131

**Published:** 2021-01-28

**Authors:** Malusi Sibiya, Mbuyu Sumbwanyambe

**Affiliations:** Department of Electrical and Mining Engineering, University of South Africa, 28 Pioneer Ave, Florida Park, Johannesburg, Roodepoort 1709, South Africa; sumbwm@unisa.ac.za

**Keywords:** VGG-16, common rust, convolutional neural networks, image histograms, fuzzy decision rules, Otsu threshold method, Image Analyzer

## Abstract

Many applications of plant pathology had been enabled by the evolution of artificial intelligence (AI). For instance, many researchers had used pre-trained convolutional neural networks (CNNs) such as the VGG-16, Inception, and Google Net to mention a few, for the classifications of plant diseases. The trend of using AI for plant disease classification has grown to such an extent that some researchers were able to use artificial intelligence to also detect their severities. The purpose of this study is to introduce a novel approach that is reliable in predicting severities of the maize common rust disease by CNN deep learning models. This was achieved by applying threshold-segmentation on images of diseased maize leaves (Common Rust disease) to extract the percentage of the diseased leaf area which was then used to derive fuzzy decision rules for the assignment of Common Rust images to their severity classes. The four severity classes were then used to train a VGG-16 network in order to automatically classify the test images of the Common Rust disease according to their classes of severity. Trained with images developed by using this proposed approach, the VGG-16 network achieved a validation accuracy of 95.63% and a testing accuracy of 89% when tested on images of the Common Rust disease among four classes of disease severity named Early stage, Middle stage, Late Stage and Healthy stage.

## 1. Introduction

Researchers who have so far used deep learning models to classify plant leaf disease severities have used training datasets that were categorized into classes that relied on human decisions by making observations. This method is biased and unreliable because some human decisions may not be accurate due to possibilities such as impaired eyesight. For instance, in a study by Sun, Wang and Wang [1], the VGG-16 was used for the classification of apple disease severities. In their study, they made use of a botanist to decide on severity stages of the diseased apple leaves. The botanist they used, made the following discretions to make decisions on severity classes: the healthy-stage leaves are free of spots; the early-stage leaves have small circular spots with diameters less than 5 mm; the middle-stage leaves have more than three spots with at least one frog-eye spot enlarging to irregular or lobed shape; the end-stage leaves are so heavily infected that they will drop from the tree. With these decisions that were made by their botanist, they developed the training datasets that were used to train the VGG-16 network with a testing accuracy of 90.4%. However, the approach they used was not generic, in the sense that the methods used to assign diseased images to their severity classes were exclusive to apple leaf diseases, and the severity class assignments were made based on the observation of a human eye, hence in this study we introduce a novel approach that uses the decisions of computerized fuzzy decision rules to assign the maize Common Rust images to their severity classes of data sets that were used to train the VGG-16 network. The proposed method is based on thresholding the images of the diseased maize (Common Rust in this case) and use the percentages of the diseased leaf area to derive fuzzy decision rules for the assignment to their severity classes. Once the severity classes were developed, we trained the fine-tuned VGG-16 network to classify the tested maize common rust images among four disease severity classes: Early Stage, Middle Stage, Late Stage, and Healthy Stage. Using this approach, the RGB images of the maize Common Rust disease were first converted to grayscale. This enabled the use of Otsu threshold-segmentation, that segmented the images into two classes of dark intensity pixels (Background) and light intensity pixels (Fore ground). To find more about the maize common rust disease, we approached the researchers of the Agricultural Research Council (ARC), South Africa. *Puccinia sorghi* is the fungus that causes maize common rust. Maize common rust disease development is favoured by conditions with cool and moist weather of around (59–72 °F). Mostly, it affects the coastal parts of Durban, South Africa. In Section 2, we review the necessary topics that made this study possible. Section 3 dwells more on the methods and materials. The results are presented in Section 4, while Section 5 discusses the findings and finally draws the conclusions.

## 2. Literature Review

The development of technology and the introduction of artificial intelligence (AI) has empowered scientists and researchers to perform plant disease detection utilizing a deep learning convolutional neural network. Deep learning is a machine subfield that has been used by researchers for many artificial intelligence applications. For instance, Zhang et al. developed an abnormal breast identification model by using a nine-layer convolutional neural network [2]. However, in the context of this study, works on deep learning for the detection of plant diseases after images have been reviewed. An Updated Faster R-CNN architecture developed by changing the parameters of a CNN model and a Faster R-CNN architecture for automatic detection of leaf spot disease (*Cercospora beticola Sacc*) in sugar beet were proposed by Ozguven and Aden [3]. Their proposed method for the detection of disease severities by the imaging-based expert systems was trained and tested with 155 images, and according to the test results, the overall correct classification rate was found to be 95.48%. A vision-based system to detect symptoms of leaf scorch on leaves of Olea europaea infected by *Xylella fastidiosa* was developed [4]. In this work, the algorithm discovered low-level features from raw data to automatically detect veins and colours that lead to symptomatic leaves. Leaf scorch was detected with a true positive rate of 98.60 ± 1.47%. The model was developed with a convolutional neural network that was trained in the stochastic gradient descent method [4]. The literature shows that the deep learning models which were used for plant disease detection had different performance accuracies that were determined by the model parameter tuning and regularization methods. Chen et al. used the pre-trained networks with Image Net, a famous dataset that has labelled images ready to be used as training data [5]. Their approach improved the performance when compared with other state-of-the-art techniques; it achieved a validation accuracy of at least 91.83% when trained on the public data set. Regardless of a complex background in the images, their proposed approach achieved a normal precision of 92.00% for the rice plant class of images [5]. Oppenheim et al. [6] used deep learning for the detection of potato tuber disease. The basic architecture chosen for this problem was a CNN developed by the Visual Geometry Group (VGG) from the University of Oxford, U.K., named CNN-F due to its faster training time [3]. In their model, several new dropout layers were added to the VGG architecture to deal with problems of over fitting, especially due to the relatively small dataset. The required input image size of their fine-tuned VGG network was a 224 × 224 matrix. The CNN comprised eight learnable layers, the first five of which were convolutional, followed by three fully connected layers and ending with a soft max layer [6]. Training CNNs usually requires a large amount of labelled data in order to perform a good classification [6]. Therefore, two methods were used for data augmentation: mirroring creates additional examples by randomly flipping the images used in training; cropping was also used, cropping the image randomly to different sizes, while keeping the cropped image’s minimum size to 190 × 190, helped to achieve data diversity [6]. Arsevonic et al. conducted a study around solving the current limitations on the use of deep learning for the detection and classification of plant diseases [7]. In their work, they used two approaches to deal with the issue of data augmentation. The first one was the augmentation by means of traditional methods, and the second one, the art style Generative Adversarial Networks (GANs) [7]. A summarized review of deep learning in plant disease classification without considering their severities is shown in Table 1.

Table 1 summarises the works of plant disease classification without considering their severities. Table 2 summarises the works that were used for the prediction of plant disease severities. The studies summarised in Table 1 and Table 2 do not in any way involve the use of threshold methods such as the Otsu method [23].

Fuzzy logic in AI provides valuable flexibility for reasoning. It is basically a method of reasoning that resembles human reasoning. This approach is similar to how humans perform decision-making, and it involves all intermediate possibilities between “YES” and “NO”. Fuzzy logic reasoning provides acceptable reasoning and helps in dealing with the uncertainty in engineering. The fuzzy logic architecture contains all the rules and the if–then conditions offered by the human experts to control the decision-making system. Behera et al. mentioned in their study that fuzzy logic was invented by Lotfi Zadeh, who observed that, unlike computers, humans have a different range of possibilities between “YES” and “NO” [29]. For this reason, Behera et al. were able to use multi class support vector machines (SVM) with K-means clustering for the classification of diseases with 90% of accuracy, and fuzzy logic to compute the degree of orange disease severity [29]. Other researchers such as Sannakki et al. proposed an image processing-based approach to automatically grade the disease spread on plant leaves by employing fuzzy logic [30]. Rastogi, Arora, and Sharma used Matlab to perform K-means based segmentation and classification, percentage infection calculation, and disease grading using the fuzzy logic toolbox [31]. The clustering method of thresholding called the Otsu method, is based on selecting a threshold value for separating the image into two classes such that the variance within each class is minimized. The selection of a threshold value modifies the spread of the two parts of the distribution while the distributions cannot be changed for obvious reasons. The key is to select a threshold that minimizes the combined spread.

The weighted sum of the variances of each class defines the within-class variance: (1)$$\sigma_{within}^{2}\left(T\right)=n_{B}\left(T\right)\sigma_{B}^{2}\left(T\right)+\;n_{F}\left(T\right)\sigma_{F}^{2}\left(T\right)$$
where:(2)$$n_{B}\left(T\right)=\;\sum_{i=0}^{T-1}p\left(i\right)$$
(3)$$n_{F}\left(T\right)=\;\sum_{i=T}^{N-1}p\left(i\right)$$
(4)$$\sigma_{B}^{2}\left(T\right)=\mathrm{Background}\;\mathrm{pixel}\;\mathrm{variance}$$
(5)$$\sigma_{F}^{2}\left(T\right)=\mathrm{Foreground}\;\mathrm{ground}\;\mathrm{pixels}$$

These above equations need a very expensive computation of the within-class variance for each class, and for each possible thresholding value, which must be avoided. To reduce the computational cost, the calculation of between-class variance that is a less expensive step can be defined as the within-class variance subtracted from the total.
(6)$$\sigma_{Between}^{2}\left(T\right)=\;\sigma^{2}-\sigma_{Within}^{2}\left(T\right)=n_{B}\left(T\right){\left[\mu_{B}\left(T\right)-\mu \right]}^{2}+\;n_{o}\left(T\right){\left[\mu_{o}\left(T\right)-\mu \right]}^{2}$$
where $\sigma^{2}$ is the combined variance and μ is the combined mean. The between-class variance is the weighted variance of the cluster means around the overall mean. By substituting $\mu =n_{B}\left(T\right)\mu_{B}\left(T\right)+n_{o}\left(T\right)\mu_{o}\left(T\right)\;$ and simplifying the result, we get
(7)$$\sigma_{Between}^{2}\left(T\right)=n_{B}\left(T\right)n_{o}\left(T\right){\left[\mu_{B}\left(T\right)-\mu_{o}\left(T\right)\right]}^{2}$$

So, for each threshold that is potential, the algorithm separates the pixels into two clusters according to the value [23]. 

## 3. Materials and Methods

In the literature, different methods of plant lea disease severity prediction had been used by other researchers [32,33,34]. Symptoms of plant diseases may include a detectable change in colour, shape, or function of the plant as it responds to the pathogen [35]. This change in colour results in the loss of the green pigment of the leaves. This proposed approach was achieved by use of the Otsu threshold-segmentation method that was used to extract the percentages of the diseased leaf area (Background pixels) which were then used to derive fuzzy decision rules for the assignment of maize Common Rust images to their severity classes. The 4 severity classes that were developed were then used to train a VGG-16 network in order to automatically classify the test images of the Common Rust disease according to their classes of severity. This enabled the VGG-16 network to make severity predictions of maize common rust disease among 3 classes, named the “Early Stage”, “Middle Stage”, and “Late Stage”. The fourth stage, named the “Healthy Stage”, was for the prediction of healthy maize leaves. For this reason, the VGG-16 was designed to be a classifier of 4 categories. The materials that were used in this study are tabulated in Table 3. Image Analyzer is basic open-source software, but amazingly effective utility for higher level analysis, editing, and optimization of images. As an open-source tool, Image Analyzer is available at https://image-analyzer.en.softonic.com/. The PlantVillage dataset has been widely used by many machine learning researchers to such an extent because it has so many online repositories. The data used in this study are available at https://github.com/spMohanty/PlantVillage-Dataset. 

To perform the Otsu thresholding on images, we used an open-source program called “Image Analyzer”. Otsu thresholding assumes that there are two classes of pixels in the image which need separating, thus an Otsu global value of 127 was used under all conditions. Figure 1 explains the procedure of our proposed approach. 

The procedure of our proposed approach is explained in Figure 1 and is illustrated in Figure 2 and Figure 3 using a maize common rust disease assigned to the Late Stage as an example.

Figure 2 and Figure 3 visually illustrate the procedure explained in Figure 1 over four steps. Step 5 is explained by means of a formula in Equation (8). The advantage of using colour image segmentation is that it is based on the colour features of the image pixels, assuming that homogeneous colours in the image correspond to separate clusters and hence meaningful objects in the image [36]. However, for the sake of simplicity, in this study, we introduce the proposed approach by first converting the images into 2-dimensional space before segmentation. We segmented the grayscale Common Rust images by use of the Otsu method at a threshold value of 127 and calculated the percentages of the diseased leaf areas by using equation (8). Then, after segmentation was complete, the dark intensities or background pixels presented the diseased leaf areas in the images while the non-diseased leaf areas, supposedly green, were presented by the light intensities or foreground pixels. The images used in this approach were of pixel size 256 × 256. Also, the images used for this approach should cover a spatial dimension of 256 pixels either in the x-dimension or y-dimension. It can be seen in Figure 2 that the spatial dimension of 256 pixels is covered by the image in the x-dimension. Finally, the background of the images should be black.
(8)$$\%\mathrm{Diseased}\;\mathrm{Leaf}\;\mathrm{Area}=\;\frac{\mathrm{Background}\;\mathrm{pixel}\;\mathrm{count}}{\left(\mathrm{Background}\;\mathrm{pixel}\;\mathrm{count}\right)\;+\;\left(\mathrm{Foreground}\;\mathrm{pixel}\;\mathrm{count}\right)\;}\;\times 100$$

According to the fuzzy decision rules that will be explained next, an image of maize common rust disease shown in Figure 2 and Figure 3 was assigned to be Late Stage because a percentage of 65.8% was calculated by means of Equation (8) as follows:$$\%Diseased\;Leaf\;Area=\frac{43175}{43175+2232}\;\times 100\;=65.8\%$$

Finally, Step 6 concerns the derivation of fuzzy decision rules for the assignment of maize common rust diseased images to their severity classes. With the help of a human expert in plant pathology, we were able to derive fuzzy decision rules for the compilation of training data sets that were categorized into “Early Stage”, “Middle Stage”, and “Late Stage”. The technique used to derive these fuzzy logic rules was based on the experience of the plant pathologist and classic methods of detecting plant disease severity, such as observing the rate at which the rust is scattered on the leaf. These rules may change depending on the plant species and the type of the disease dealt with. As for the “Healthy Stage”, the training data were compiled by use of healthy images in the PlantVillage dataset. 

The compilation of the training data sets was conducted using fuzzy decision rules outlined subsequently.

Design Rules for Healthy Prediction.

Rule 1: As for the “Healthy Stage”, the training data were compiled by use of healthy images in the PlantVillage dataset.

Design Rules for Common Rust Disease Severity Prediction.

Fuzzy decision Rule 1: If %Diseased Leaf Area ≥50%, then, the image belongs to “Late-stage training data set”.

Fuzzy decision Rule 2: If 45% ≤ %Diseased Leaf Area < 50%, then, the image belongs to “Middle stage training data set”.

Fuzzy decision Rule 3: If %Diseased Leaf Area <45%, then, the image belongs to “Early-stage training data set”.

Figure 4 illustrates the procedure for the assignment of maize common rust diseased images to their severity classes using the fuzzy decision rules. The same procedure was repeated for all the common rust images until enough datasets were compiled to train the fine-tuned VGG-16 network. There were three severity classes of maize common rust disease and one healthy class. The fourth healthy class was developed from the PlantVillage dataset by use of healthy maize images (Appendix A).

Figure 5 shows a sample of healthy leaf images from the PlantVillage datasets that were used for training in the fourth class.

By performing the above fuzzy decision rules to assign the maize common rust diseased images to their severity classes, and formation of a healthy class with healthy images of the PlantVillage dataset, we ended up with four categories of the training datasets. Figure 6 shows the arrangement of the datasets which were adhered to in order to be able to train the VGG-16 network with the images of our proposed approach.

We noticed that the images in the PlantVillage dataset were taken in almost similar daylight conditions. To be certain of the VGG-16’s performance trained in the dataset proposed in this study, we used an A-30 Samsung 16-megapixel camera under different light conditions. The images were taken in summer, in average South African weather conditions. The first set of images was taken at 05:00 to 06:00 a.m. in the rising sun. The second were taken around midday, while the last were taken in the evening hours of sunset at about 18:00–19:00 p.m. During testing, equal images were used for different light conditions.

### 3.1. Development of a Deep Learning Model by Fine-Tuning a VGG-16 Network

#### 3.1.1. Theoretical Background of the VGG-16 Network

The VGG-16 network was proposed by Simonyan and Zisserman of the Visual Geometry Group Lab of Oxford University in 2014 [37]. This model took the first and second prize in the ImageNet Large Scale Visual Recognition competition on categories of object localization and image classification, respectively. There were 200 classes in the object localization category, and 1000 classes in the image classification category. The architecture of the VGG-16 is illustrated in Figure 7.

It can be seen in Figure 7 that the VGG-16 network has an input tensor of (224,224,3). This model processes the input image and outputs a vector of 1000 prediction values (probability) as shown in Equation (9).
(9)$$\hat{y}=\;\left[\begin{array}{c} {\hat{y}}_{0}\\ {\hat{y}}_{1}\\ .\\ {\hat{y}}_{999} \end{array}\right]$$

The classification probability for a corresponding class is determined by a Softmax function as shown in Equation (10). Equation (10) shows the prediction probability for the *j*th class given a sample vector *X* and the weighting vector *W* using a Softmax function.
(10)$$P(y=j|X)=\;\frac{e^{X_{W_{j}}^{T}}}{\sum_{k=1}^{K}e^{X_{W_{k}}^{T}}}$$

#### 3.1.2. Fine-tuning and Training a VGG-16 Network for Maize Leaf Disease Severity Prediction

There are four scenarios in which a pre-trained model can be fine-tuned. These scenarios are summarized as follows:

Scenario 1: The target dataset is small and quite similar to the source dataset.

Scenario 2: The target dataset is large and quite similar to the source dataset. 

Scenario 3: The target dataset is small and very different from the source dataset

Scenario 4: The target dataset is large and very different from the source dataset.

The guidelines for the appropriate fine-tuning level to use in each of the scenarios are summarised in Table 4.

Our model for predicting the severities of the maize common rust disease was developed by fine-tuning a VGG-16 network as guided by scenario 3 of Table 4. This was achieved by freezing all the layers of a VGG-16 network, except for the top four layers. The fine-tuned model that resulted is summarised in Table 5.

Our model had a total of 15,517,668 parameters, of which 802,980 of them were trainable, while 14,714,688 were not. It can be seen in Table 5 that our FC (fully connected) layer consisted of a dense layer with 32 nodes and an activation function of “relu”. The dropout layer was also used to randomly switch off 20% of the nodes in the first dense layer of our FC, during each training epoch. The output layer consisted of a dense layer with four nodes and “Softmax” as an activation function. Each class of the four classes was trained with 400 images and validated with 50 images. This means that the training data consisted of 1600 images while a total 200 images were used for validation. With a batch size of 32, the training steps per epoch were determined by a number of training samples divided by a batch size, which in this case was 1600/32. To determine the validation steps per epoch, we also divided the total number of validation samples by a batch size, which in this case was 200/32. “Adam” was the choice of the optimizer we used, that enabled the model to result in a high validation accuracy of 95.63 % and a validation loss of 0.2 when the learning rate was set to 0.0001.

## 4. Results

Table 6 summarises the information of the datasets, tuned hyperparameters, and the performance metrics that defined our fine-tuned VGG-16 network for the prediction of maize common rust disease severities.

Figure 8 and Figure 9 show the plots of training metrics against validation metrics. Figure 8 shows the loss metrics plots, and Figure 9 shows the accuracy metrics plots. The main cause of poor prediction performance in machine learning is either overfitting or underfitting the data [38]. Underfitting means that the model is too simple and fails to fit (learn) the training data [38]. Overfitting means that the model is so complex that it memorizes the training data and fails to generalize from test/validation data that it has not seen before [38]. Therefore, it can be seen in Figure 8 and Figure 9 that the proposed model did not underfit nor overfit the training data, because both plots show that they are generalized well from the validation data. The model also achieved a high testing accuracy of 89% when it was tested against 100 images, with 25 images in each class. It can also be seen in Figure 8 that the validation loss converged to 0.2 without oscillations, which was positive, and indicated that a learning rate of 0.0001 that we set in the Adam optimizer was ideal.

The testing dataset were a collection of camera images that were also assigned to their severity classes using the approach proposed in this study. A total of 100 images were used for testing. Out of the chosen 100 images for testing, each class contained 25 images. Figure 10 shows the number of correct classifications that the VGG-16 was able to make in each class. Equation (11) shows how the testing accuracy of 89% was achieved by the VGG-16 network.
(11)$$\mathrm{Testing}\;\mathrm{Accuracy}=\;\frac{\mathrm{Number}\;\mathrm{of}\;\mathrm{correct}\;\mathrm{classifications}\;\mathrm{in}\;\mathrm{each}\;\mathrm{class}}{\mathrm{Total}\;\mathrm{number}\;\mathrm{of}\;\mathrm{testing}\;\mathrm{images}}\;\times \;100$$

Using the information provided in Figure 10, the model’s testing accuracy was calculated as follows.
$$\mathrm{Testing}\;\mathrm{Accuracy}=\frac{23+21+22+23}{100}\;\times \;100=89\%$$

## 5. Discussion and Conclusions

Considering the works on plant leaf disease severity prediction by deep learning models, we introduce a novel approach of assigning the diseased plant leaf images (maize common rust in our case) to their severity classes guided by the fuzzy decision rules that were derived from the calculated percentages of the diseased leaf areas. To accomplish this, we first converted the colour images from a 3-dimesional array to a 2-dimensional array. A 3-dimensional array has three channels of red, green and yellow colours. Each of these channels is made up of 8-bit pixels that determine the colour intensity in different parts of the image. For instance, a green colour is a result of setting the intensities of the pixels in the same dimensional space to 255G + 0B + 0R. The approach proposed in this study uses segmentation, which is in fact tedious when performed in the 3-dimesional space. The best way to achieve our goals was to first convert colour images from a 3-dimensional space to a 2-dimensional space. Figure 11 shows the differences between two colour spaces that the image can take. Next, we segmented the grayscale images and calculated the percentages of the diseased leaf areas in the maize Common Rust images. This enabled us to create fuzzy decision rules as guided by an experienced plant pathologist. For instance, he mentioned that a dark common rust which covers more than 50% of the leaf area is categorized as a late-stage disease. These fuzzy logic rules were then used to develop a training dataset that was used train the VGG-16 network. The maize common rust severity classes developed in this way were used to train the fine-tuned VGG-16 that obtained a validation accuracy of 95.63% and a testing accuracy of 89%.

The approach that was proposed by Wang, Sun, and Wang [1] for plant leaf disease severity prediction using Deep Learning was biased in a way because the methods they used to assign the images of the Common Rust disease to their severity classes were totally based on decisions that were made by human eye observations. The proposed approach is therefore unbiased as it utilizes the decisions of computerized fuzzy decision rules to assign the Common Rust images to their severity classes for maize Common Rust disease. To the best of our knowledge, this is the first report of the approach to predict the severity of maize common rust disease. Broadly translated, our findings indicate that the fine-tuned model (VGG-16) trained with the datasets compiled in this study’s approach has higher validation and testing accuracies in the prediction of maize common rust disease severity. Future investigations are necessary to validate the kinds of conclusions that can be drawn from this study if other types of maize leaf diseases, or any other leaf disease types of different plant species were to be used in this study’s approach. 

## Figures and Tables

**Figure 1 pathogens-10-00131-f001:**
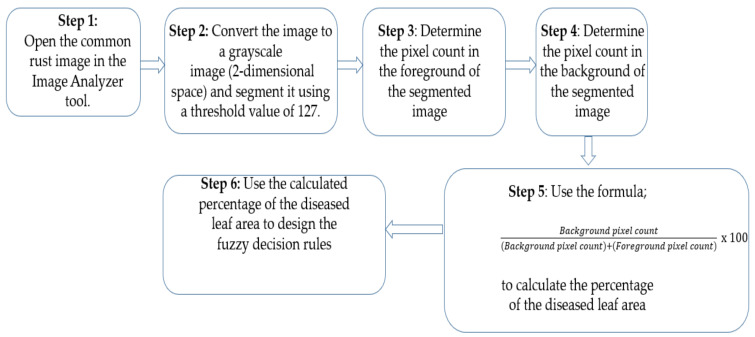
The procedure of the proposed approach.

**Figure 2 pathogens-10-00131-f002:**
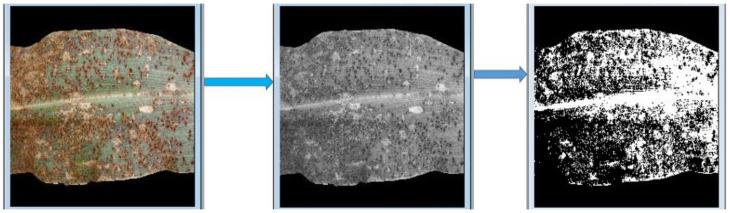
Procedure explained in Figure 1.

**Figure 3 pathogens-10-00131-f003:**
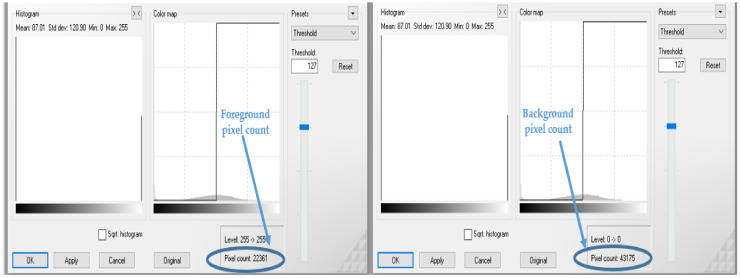
Procedure explained in Figure 1, continued.

**Figure 4 pathogens-10-00131-f004:**
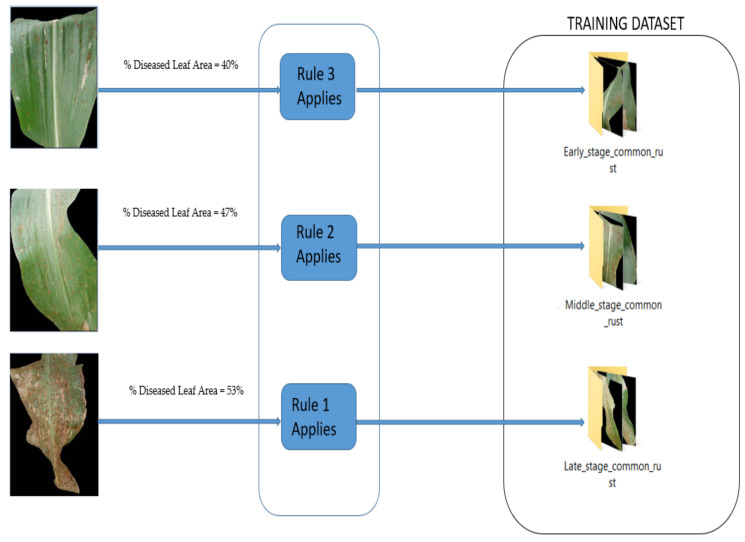
Assignment of maize common rust disease images to their severity classes using fuzzy decision rules.

**Figure 5 pathogens-10-00131-f005:**
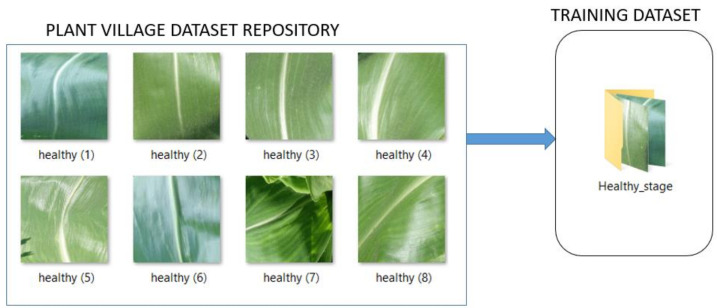
A sample of healthy maize images that were used for training in the Healthy class.

**Figure 6 pathogens-10-00131-f006:**
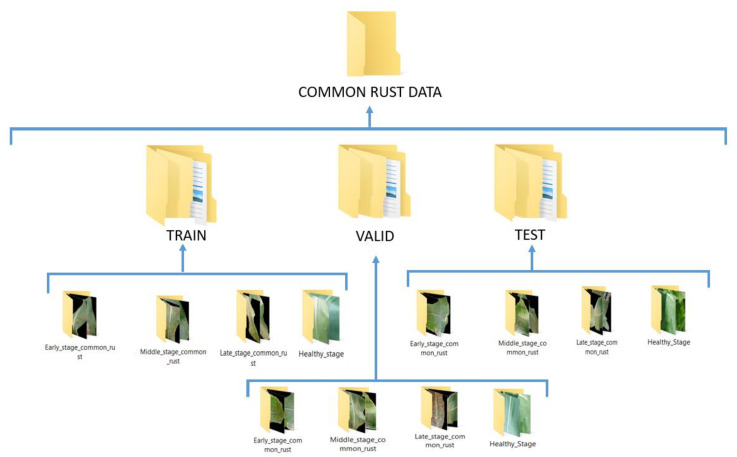
Final arrangement of the training, validation, and test data sets for the prediction of the maize common rust disease severities by a fine-tuned VGG-16 network.

**Figure 7 pathogens-10-00131-f007:**
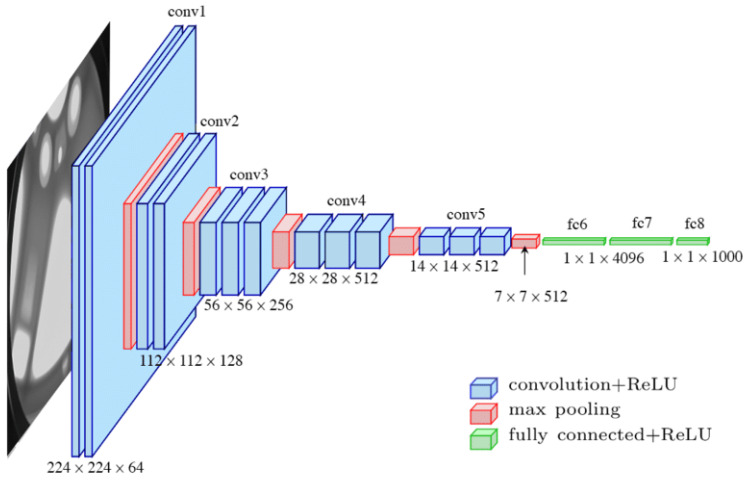
The VGG-16 architecture.

**Figure 8 pathogens-10-00131-f008:**
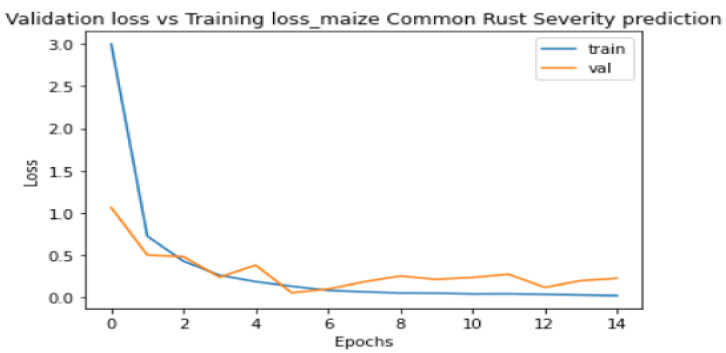
Training loss against validation loss plots.

**Figure 9 pathogens-10-00131-f009:**
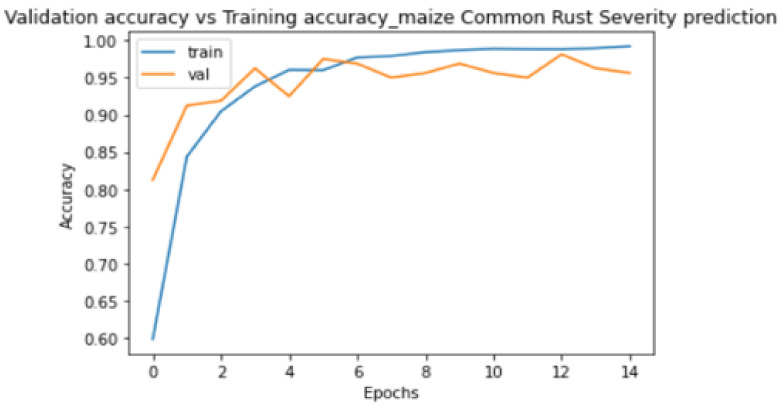
Training accuracy against validation accuracy plots.

**Figure 10 pathogens-10-00131-f010:**
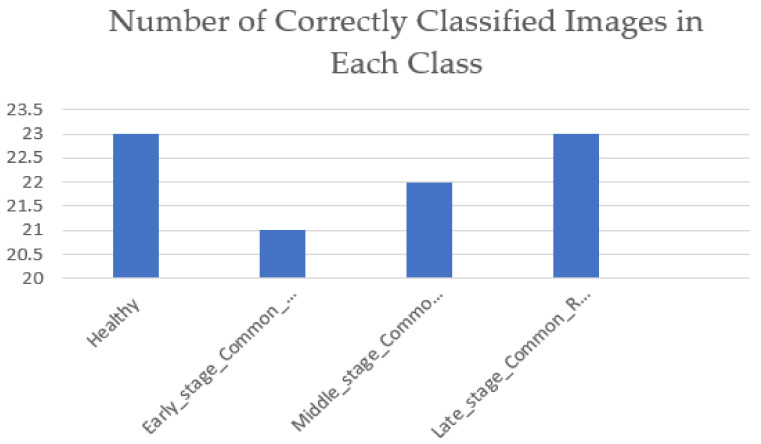
Number of correctly classified images in each class by the VGG-16 network.

**Figure 11 pathogens-10-00131-f011:**
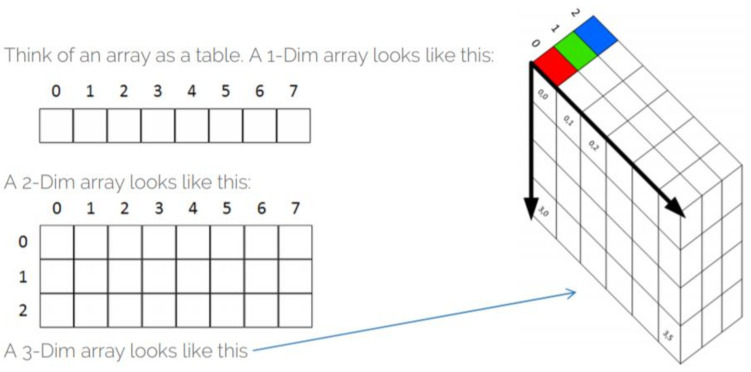
A comparison of 2-dimensional array images with 3-dimensional array images.

**Table 1 pathogens-10-00131-t001:** A summary of deep learning models used in plant disease classification.

Deep Learning Architecture/Image Database	Study Summary
Google Net	In this study, 12 plant species were used from a known image database, with each plant species having a different number of samples and diseases under a variety of conditions [8].
U-Net	After using the U-Net to obtain images of the segmented leaves, the next task was to identify the plant disease types. This is a typical image classification task [9].
3D Deep Convolutional Neural Network (DCNN)	Based on hyperspectral imaging of inoculated and mock-inoculated stem images, the 3D DCNN achieved a classification accuracy of 95.73% and an infected class F1 score of 0.87 [10].
ImageNet	Brahimi, Boukhalfa, and Moussaoui used the 1000 class ImageNet database with a pretrained model to classify 9 types of tomato diseases [11].
Dense Net	A lightweight deep neural network (DNN) approach that can run on internet of things (IoT) devices with constrained resources was proposed [12].
9-Layer Deep CNN	The deep CNN model was trained using an open dataset with 39 different classes of plant leaves and background images. Six types of data augmentation methods were used: image flipping, gamma correction, noise injection, principal component analysis (PCA) colour augmentation, rotation, and scaling. The proposed model achieved 96.46% classification accuracy [13].
Deep Siamese convolutional network	The deep Siamese convolutional network was developed to solve the problem of the small image databases. Accuracy over 90% was reached in the detection of the Esca, Black rot and Chlorosis diseases on grape leaves [14].
CNN API written in Python	The model was designed to detect and recognize several plant varieties, specifically apple, corn, grapes, potato, sugarcane, and tomato. The trained model achieved an accuracy rate of 96.5%, and the system was able to register up to 100% accuracy in detecting and recognizing the plant variety and the type of diseases with which the plant was infected [15].
Alex Net and Google Net	Using a public dataset of 54,306 images of diseased and healthy plant leaves collected under controlled conditions, a deep convolutional neural network was trained to identify 14 crop species and 26 diseases (or absence thereof). The trained model achieved an accuracy of 99.35% on a held-out test set, demonstrating the feasibility of this approach [16].
Alex Net, VGG16, and VGG19	The experiments were carried out using data consisting of the real disease and pest images from Turkey. The accuracy, sensitivity, specificity, and F1-score were all calculated for performance evaluation [17]
Alex Net, Alex Net OWTBn, Google Net, Over feta and VGG	Convolutional neural network models were developed to perform classifications of plant disease using simple leaf images of healthy and diseased plants, through deep learning methodologies. Training of the models was performed with the use of an open database of 87,848 images, containing 25 different plants in a set of 58 distinct classes of [plant, disease] combinations, including healthy plants. Several model architectures were trained, with the best performance reaching a 99.53% success rate in identifying the corresponding [plant, disease] combination (or healthy plant) [18].
CNN model similar to standard Le Net architecture	Images of apple leaves, covering various diseases as well as healthy samples, from the PlantVillage dataset were used to validate results. Image filtering, image compression, and image generation techniques were used to gain a large training set of images and tune the system perfectly. The trained model achieved high accuracy scores in all the classes, with a net accuracy of 98.54% on the entire dataset, sampled and generated from 2561 labelled images [19].
Alex Net and VGG-16	In this study tomato-leaf images (6 diseases and a healthy class) obtained from the PlantVillage dataset were provided as inputs to two deep learning-based architectures, namely, Alex Net and VGG16 net. The accuracy of classification obtained using Alex Net and VGG16 net were 97.49% and 97.23%, respectively, for 13262 images [20].
VGG 16, Inception V4, Res Net with 50, 101 and 152 layers and Dense Nets with 121 layers	An empirical comparison of the Deep Learning architecture was done. The architectures evaluated included VGG 16, Inception V4, Res Net with 50, 101 and 152 layers and Dense Nets with 121 layers. The data used for the experiment were 38 different classes, including diseased and healthy leaf-images of 14 plants from PlantVillage. Fast and accurate models for plant disease identification were desired, so that accurate measures could be applied early. In the experiment, Dense Nets had a tendency to consistently improve in accuracy with the growing number of epochs, with no signs of overfitting and performance deterioration. Moreover, Dense Nets required a considerably less number of parameters and reasonable computing time to achieve state-of- the-art performances. It achieved a testing accuracy score of 99.75% to beat the rest of the architectures [21].
Google Net	The work explored the use of individual lesions and spots for the task, rather than considering the entire leaf. Each region had its own characteristics, therefore the variability of the data was increased without the need for additional images. This also allowed the identification of multiple diseases affecting the same leaf. On the other hand, suitable symptom segmentation still needed to be performed manually, preventing full automation. The accuracies obtained using this approach were, on average, 12% higher than those achieved using the original images. Additionally, no crop had accuracies below 75%, even when as many as 10 diseases were considered. Although the database did not cover the entire range of practical possibilities, these results indicated that, as long as enough data were available, the deep learning technique is effective for plant disease detection and recognition [22].

**Table 2 pathogens-10-00131-t002:** A summary of methods for plant disease severity detection.

Methods Used for Plant Disease Severity Prediction	Study Summary
Using deep learning to automatically to automatically predict plant disease severity.	Wang, Sun and Wang developed different deep learning models that also included the VGG-16 to detect plant severities in four stages. The VGG-16 surpassed the other models with a validation accuracy of 90.4% [1].
Use of image processing to measure leaf disease symptoms.	In this study, the segmentation of the leaf area was performed by a simple threshold method, and that of the lesion region area with the triangle threshold method. The quotient of lesion area and leaf area were used to categorize the diseases with testing accuracy of 98.60% [24].
Use of digital image processing to measure leaf disease symptoms.	Barbedo also used image processing and measurements to detect severities of plant leaf diseases. Using his method, he received an accuracy of 96% to detect the severities of the plant leaf diseases he dealt with [25]. However, his method also has a disadvantage of limiting users who are not scientists or familiar with image processing.
Plant disease incidence and severity measurements by use of machine learning.	Owomugisha and Mwebaze used machine learning algorithms such as support vector machines and k nearest neighbours to detect the plant disease incidents and severity measurements. In their work, they used traditional machine learning algorithms that required handcrafted feature extraction. The handcrafted feature extraction algorithms they used for colour extraction in the images were SURF, SIFT, OBR and HOG [26].
Segmentation of the affected area	The proposed work used two cascaded classifiers. Using local statistical features, the first classifier segmented the leaf from the background. Then, using hue and luminance from the Hue-Saturation-Value colour space, another classifier was trained to detect disease and its stage of severity [27].
Image segmentation and colour sensing	Various new parameters, namely disease severity index (DSI), infection-per region (IPR), and disease-level parameter (DLP) for measuring the disease severity level and level-classification were formulated and derived [28].

**Table 3 pathogens-10-00131-t003:** A tabulated summary of the materials that were used in the study.

Hardware	Software	Dataset
ASUS TUF Laptop with NVIDIA GeForce GTX 1650 4GB Graphics. Samsung A-30 16 Mega Pixels rear camera	Image Analyzer, an open-source image processing tool. Jupyter Notebook IDE. Python-Keras library. Python-Tensor flow library. Anaconda package.	A standard PlantVillage maize data set.

**Table 4 pathogens-10-00131-t004:** Summary for the fine-tuning of pretrained models.

Scenario	Size of the Target Data	Similarity of the New and Original Data Sets	The Approach Used
1	Small	Similar	The pre-trained network is used as feature extractor.
2	Large	Similar	Fine-tuning is conducted through the full network.
3	Small	Very different	Fine-tuning is conducted from activations earlier in the network.
4	Large	Very different	Fine-tuning is conducted through the entire network.

**Table 5 pathogens-10-00131-t005:** Model summary for a fine-tuned VGG-16 network to predict maize common rust disease severities.

Layer Name	Type	Number of Filters	Number of Parameters
Input	Input layer	-	0
Block 1_Conv2D_1	Convolutional	64	1792
Block 1_Conv2D_2	Convolutional	64	36,928
Block 1_MaxPooling2D	Max Pooling	-	0
Block 2_Conv2D_1	Convolutional	128	73,856
Block 2_Conv2D_2	Convolutional	128	147,584
Block 2_MaxPooling2D	Max Pooling	-	0
Block 3_Conv2D_1	Convolutional	256	295,168
Block 3_Conv2D_2	Convolutional	256	590,080
Block 3_Conv2D_3	Convolutional	256	590,080
Block 3_MaxPooling2D	Max Pooling	-	0
Block 4_Conv2D_1	Convolutional	512	1,180,160
Block 4_Conv2D_2	Convolutional	512	2,359,808
Block 4_Conv2D_3	Convolutional	512	2,359,808
Block 4_MaxPooling2D	Max Pooling	-	0
Block 5_Conv2D_1	Convolutional	512	2,359,808
Block 5_Conv2D_2	Convolutional	512	2,359,808
Block 5_Conv2D_3	Convolutional	512	2,359,808
Block 5_MaxPooling2D	Max Pooling	-	0
Flatten	Layer	-	0
Dense (32 nodes)	Layer	-	802,848
Dropout (0.2)	Layer	-	0
Dense_1 (4 nodes)	Layer	-	132

**Table 6 pathogens-10-00131-t006:** Summary of model hyper parameter tuning and performance metrics.

Total Training Images.	1600
Total Validation Images.	200
Total images from the collected dataset.	100
Batch size.	32
Optimizer Type and Learning rate.	Adam (lr = 0.0001)
Dropout.	20% of nodes in the first Dense layer of FC.
Training Accuracy on the PlantVillage dataset.	99.21%
Validation Accuracy on the PlantVillage dataset.	95.63%
Test Accuracy on the collected data set.	89%
Training Epochs.	15
Training Loss.	0.02
Validation Loss.	0.2

## Data Availability

The training data set is available on request by sending an email to 43447619@mylife.unisa.ac.za.

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
