# Peer review of "Automatic Fuzzy Logic-Based Maize Common Rust Disease Severity Predictions with Thresholding and Deep Learning"

_pathogens, 2021, doi:10.3390/pathogens10020131_

Round 1

Reviewer 1 Report

Title:

Automatic Fuzzy Logic-Based Maize Leaf Disease 2 Severity Predictions with Thresholding and Deep 3 Learning

In this work the authors use a pre-trained Convolutional Neural Networks (CNNs) for analyze severities of leaf diseases applied to maize affected by common Rust  extracting  the percentages of the green pixels to derive fuzzy decision rules, for to connect of the diseased images to their severity classes.

The work is interesting and innovative. In the introducion section the authors present a  long list of methods used in agriculture to evaluate diseases in the field using image database included in a big table format.  This method is not  conventional for a research article but useful for understanding the systems shown. 

In the paper the survey method and the programs used are described, however, for a journal like this it would be appropriate to give also some information about  the disease analyzed in this study (the casual agent of disease, which organs of the plant are affects, only the leaves? Which cultivar/s were analyzed. Authors must explain whether evaluating leaf symptoms is a direct index of disease severity in the field.

Here are some suggestions.

Line 153 in the Materials and Methods section add a short paragraph for describe the plant material used for this study, when and where it was sampled in the field, how many leaf images were analyzed to set up the system. Add also some informations about the Common Rust disease (see above).  I think these informations are important to understand the usefulness of the proposed approach.

The table 4 could be organize as flux diagram imagine of the pipeline.

Line 196: ‘With the help of a human expert in plant pathology’

Specify better what he did

Figures 4,5,6 7, 8

The images quality is poor. Improve it.

Table 7 – organize better you can put the parameters in the vertical column and the data in horizontal order to save some space and make the table more readable

The discussion is very poor. In the introduction you add a list of lot similar approaches but in this section you can better explain the differences of your method  with the others used. Also you can add some infromation about the impact of this methods on common Rust  disease control. The

Supplemental Materials must be indicated in the text

Check the references section. There are many formal inaccuracies

Author Response

The comments and advices were very fruitful and we revised the paper according to each of these comments.Thanks a lot

Reviewer 2 Report

A topic of the manuscript is interesting and undoubtedly important. The techniques for quick and reliable diagnostics of fungal plant diseases are very important for the plant breeders, farmers and plant protection specialists. However, from the duty of the reviewer, I must point out some defects of the manuscript.

The title: The title should correspond exactly to the topic of the manuscript. So, is MS about maize leaf disease or just about common rust of maize?

Keywords: Again, as above. Leaf diseases or precisely common rust? Furthermore, please avoid any abbreviations in keywords (CNN).

Introduction: In the Introduction there is no information about the pathogen. The full Latin name and some information about the presence and environmental requirements of the pathogen and its economic significance should be given. Not all readers from different countries well know the disease being examined.

Lines 45-6 & 52: "Diseased images" is imprecise term. What is diseased: an image or a plant?

Literature review:

Lines 58-9: All abbreviations used in MS for the first time should be explained.

Lines 65 & 69: The Latin names should be italicized.

Lines 73 & 81: Should be “Deep Learning” written normally or in capitals? Please decide.

Lines 75 & 81: Please insert the reference numbers directly after the authors.

Table 1: Please remove the blank row.

Line 106: Please use here the abbreviation AI. But this abbreviation should be previously explained - see line 58.

Line 113: The reference [29] refers to the publication by Behera et al. not to publication by Lotfi Zadeh. And please explain the abbreviation SVM.

Line 153: Any reference related to Otsu threshold method would be expected.

Table 3: Image Analyzer program is mentioned in this place for the first time. Hence the appropriate reference or URL to the program would be expected. Furthermore, the term "Plant Village data set" should be specified. Please don't forget that not all readers from different countries will know what Plant Village means and what data does this institution make available. Or simply give the URL.

Table 4: The following information would be expected at somewhere: how was the digital image acquisition made? How were the leaves cut off? Were the symptoms of the disease measured at the top, middle or bottom (closer to the stalk) part of the leaf? In the whole MS there is no information about the imaging device (camera? Smartphone? Scanner?), resolution, light source etc. If I understand all images come originally from Plant Village, but in the manuscript there should be any basic information about acquisition conditions. The publication concerns the use of the image analysis. If the described concept is not only theoretical but also practical, the appropriate information about the whole process of acquisition is necessary to repeat the similar studies in different sites. Furthermore, please explain if 127 a "general" value or does it relate to the specific conditions of described experiment?

Figure 2 (caption): The threshold procedure is not clearly described. If I understand correctly, the source images were 24-bit RGB. If the 127 threshold is applied to such an image, the threshold will be valid to all three color components in the same degree. This means you cut the values for R, G and B at 127 and after the segmentation the minimum values will be 0R, 0G and 0B. The maximum values (255) are then maintained. If my thinking is correct, then (1) Why did the Authors apply the threshold at 127 for the green component for the second time? Wouldn't it be easier to simply split the source color image into 3 channels (RGB) and to perform further procedures on an 8-bit G-image? (2) The phrase "to calculate the percentages of the green colour in each image..." is in my opinion incorrect. What is "green color"? Either we are thinking only about the G-values of the individual image pixels or the specific G-value relationships of each RGB pixel to the two components R and B. (3) It is not quite correct to use JPG images for color analysis (because JPG is a loss compression), especially with such small images of 256x256 pixels. But let it be.

 Figure 5 (caption): Each scheme is characterized by some type of simplification but (1) The images shown in the scheme are identical for the train, valid and test series. Does this suggest anything? (2) Is it fully correct, in the Authors opinion, to use the same images in the train and test series? (see attached Supplementary materials). Although the following sections of the manuscript describe different scenarios (table 5) I think the Authors should comment on this somewhere sooner.

Lines 333-4: In my opinion this URL should be moved to the section References.

References: The style of References should be adjusted strictly to the requirements of the journal.

Author Response

Thank you to reviewer 2 for his input .He has brought quality to the manuscript.His comments and ability to notice small  important  things is very much appreciated.

Round 2

Reviewer 2 Report

The manuscript has been substantially improved. I thank the Authors for making such significant changes in so short time. In my opinion the current version of MS can be published "as is" in the journal Pathogens.